# Real-Time Cinematic Tracking of Targets in Dynamic Environments

Ludovic Burg

Univ Rennes, Inria, CNRS, IRISA
France

Christophe Lino

LIX, École Polytechnique, CNRS, IP Paris
Palaiseau, France

Marc Christie*

Univ Rennes, Inria, CNRS, IRISA
France

## ABSTRACT

Tracking in a cinematic way a moving target inside a 3D dynamic environment remains a challenging problem. This requires to simultaneously ensure a low computational cost, a good degree of reactivity and a high cinematic quality despite sudden changes. In this paper, we draw on the idea of Motion-Predictive Control to propose an efficient real-time camera tracking technique which ensures these properties. Our approach relies on the predicted motion of a target to create and evaluate a very large number of camera motions using hardware ray casting. Our evaluation of camera motions includes a range of cinematic properties such as distance to target, visibility, collision, smoothness and jitter. Experiments are conducted to display the benefits of the approach with relation to prior work.

## 1 INTRODUCTION

The automated generation of qualitative camera motions in 3D virtual environments is a key problem for a number of computer graphics applications such as computer games, automated generation of virtual tours or virtual storytelling. The first and foremost problem is to identify what are the intrinsic characteristics of good camera motions. While film literature provides a thorough and in-depth analysis of what makes a qualitative viewpoint in terms of framing, angle to target, aesthetic composition, depth-of-field or lighting, the characterisation of camera motions has been far less addressed. This pertains to specifics of real camera rigs (dollies, cranes) that physically limit the range of motions, and also the limited use of long camera sequences in movies with the exception of Steadicam sequence shots. In addition, characteristics of camera motions in movies are strongly guided by the narrative intentions which need to be conveyed (*e.g.* rhythm, excitation, or soothing atmosphere) that are difficult to formalize.

In an attempt to transpose this knowledge to the tracking of targets in virtual environments, one can however derive a number of desirable cinematic characteristics such as *visibility* (avoiding occlusion of the tracked target and collisions with the environment), *smoothness* (avoiding jerkiness in trajectories) and *continuity* (avoiding large changes in viewing angles and distances to target). In practice, however, these characteristics are often contradictory: avoiding a sudden occlusion requires a strong acceleration, or an abrupt change in angle. Furthermore, the computational cost of evaluating visibility, continuity and smoothness along trajectories limits the possibility of evaluating many alternative camera motions.

Existing work have either addressed the problem using global motion planning techniques typically based on precomputed roadmaps [5, 10, 11], or local planning techniques using ray casting [12] and shadow maps for efficient visibility computations [1, 2, 4]. While global motion planning techniques excel at ensuring visibility given their full prior knowledge of the scene, local planning techniques excel in handling strong dynamic changes in the

---

*e-mail:marc.christie@irisa.fr

environment. The main bottleneck of both approaches remains the limited capacity in evaluating at run time the cinematic properties along a camera motion or in the local neighborhood of a camera position.

Our approach builds on the idea of performing a mixed local+global approach by exploiting a finite-time horizon that is large enough to perform a global planning, yet efficient enough to react in real-time to sudden changes. This sliding window exploits recent hardware raycasting techniques to enable the real-time evaluation of thousands of camera motions. As such, our approach draws inspiration from Motion-Predictive Control techniques [13] by optimizing a finite time-horizon, only implementing the current timeslot and then repeating the process on the following time slots.

To implement this approach, we make the hypothesis that the target object is controlled by the user through interactive inputs. Its motions and actions can therefore be predicted within a short time horizon $h$. Our system comprises 2 main stages, illustrated in Figure 1. In the first stage, we predict the motion of the target over our given time horizon $h$ by using the target's current position (at time $t_i$) and the user inputs. We then select an ideal camera position at time $t_i + h$ and propose to define a *camera animation space* as a collection of smooth camera animations that link the current camera position (at time $t_i$), to the ideal camera location (at time $t_i + h$). In the second stage, we perform an evaluation of the quality of the camera animations in this animation space by relying on hardware raycasting techniques and select the best camera animation. In a way similar to Motion-Predictive Control [13], we then apply part of the camera animation and re-start the process at a low frequency (4 Hz) or when a change in the user inputs is detected. Finally, to better adapt the camera animation space to the scene topology (*e.g.* cluttered environments *vs.* open environments), we dynamically update a scaling factor on the animation space. As a whole this process allows generating a continuous and smooth camera animation which enables the real-time tracking of a target object in fully dynamic and complex environments.

Our contributions are:

- the design of a camera animation space as a finite time horizon space in which to express a range of camera trajectories;
- an efficient evaluation technique using hardware ray casting;
- a motion predictive control approach that exploits the camera animation space to generate real-time cinematic camera motions.

## 2 RELATED WORK

We narrow the scope of related work to real-time camera planning techniques. For a broader view of camera control techniques in computer graphics, we refer the reader to [3].

### Global camera planning

Global camera path-planning techniques build on well-known results from robotics such as probabilistic roadmaps, regular cell decomposition or Delaunay triangulation. All have in common the computation of a roadmap as a graph where nodes represent regions of the configuration-free space (points, regular cells or other primitives), and edges represent collision-free links between the nodes. Niewenhuisen and Overmars exploited probabilistic roadmaps (PRM) to

automatically perform camera path queries within the graph structure, to link given starting and ending camera configurations [10]. Different heuristics were used to smooth the camera trajectories and avoid sudden changes in position and camera angles. Later, Oskam *et al.* [11] proposed a visibility-aware roadmap by using a sphere-sampling of the configuration-free space, and by precomputing the sphere-to-sphere visibility using stochastic ray-casting.

Lino *et al.* [7] exploited spatial partitioning techniques to compute dynamically evolving volumes around targets. The adjacency between the volumes was then exploited to dynamically create a roadmap through which camera paths were computed while accounting for visibility and viewpoint semantics along the path. More recently, Jovane *et al.* [5] exploited the topological representations of 3D environments to create topology-aware camera roadmaps that lower the complexity (compared to probablistic roadmaps) and enable the generation of different cinematic styles.

Yet, the cost of precomputing the roadmap together with the difficulty in dynamically updating it to account for changes limits the practical applicability of such techniques in strongly dynamic environments such as those met in computer games or storytelling applications.

**Local camera planning**

Local camera planning techniques rely on a limited knowledge of the environment. By sampling the local neighborhood around the current camera location, such systems are able to take decisions as to were to move the camera next. The decision is guided by the thorough evaluation of cinematic properties such as visibility, smoothness and continuity on the camera samples. To reduce the computational cost in the evaluation of target's visibility, Halper *et al.* [4] exploit hardware rendered shadow maps that compute potential visible sets. Results are coupled with a hierarchical solver to handle other cinematic properties. Later, Normand and Christie exploited slanted rendering frustums to compose spatial and temporal visibility for two targets over a small temporal window (10 frames) [2]. Additional criteria were added in order to select the best move to perform at each frame and to balance between camera smoothness and camera reactivity. Litteneker *et al.* [8] proposed a local planning technique based on an active contour algorithm.

Hardware rendering was also exploited by Burg *et al.* [1] who performed shadow map projections from the targets to the surface of the Toric manifold (a specific manifold space dedicated to camera control [6]). The visibility information provided by the shadow maps was then exploited to move the camera on the surface of the Toric manifold while ensuring secondary visual properties.

Recently, for the specific case of drone cinematography, Nageli *et al.* [9] built a non-linear model predictive contouring controller to jointly optimize 3D motion paths, the associated velocities and control inputs for an autonomous drone.

Our approach partly builds on the work of Nageli *et al.* , by borrowing the idea of a receding horizon process in which motion planning is performed for a large enough time horizon (few seconds). The processed is repeated at a higher frequency to account for dynamic changes in the environment. Rather than addressing the problem using a non-linear solver, we propose in our paper to exploit the hardware raycasting capacities of recent graphics cards to efficiently detect collisions and occlusion and evaluate visual properties of thousands of camera trajectories for each time horizon.

## 3 OVERVIEW

Our system aims at tracking in real-time a target object traveling through a dynamic 3d environment by generating series of smooth cinematic camera motions. In the following, we will present the construction of our camera animation space (Section 4) and then detail the evaluation of camera animations using hardware ray casting (Section 5). We will then show how the camera animation space

| | |
|---|---|
| $H^i$ | Time horizon for iteration $i$ (between times $t_i$ and $t_i + h$) |
| $\mathbf{B}^i(t)$ | Target behavior (predicted position) at time $t \in H_i$ |
| $\mathbf{V}^i$ | Set of preferred viewpoints at time $t_i + h$ |
| $\mathbb{Q}^i$ | Camera *animation space* for horizon $H^i$ |
| $M^i$ | Transform matrix of the camera animation space, for $H^i$ |
| $\mathbf{q}^i_j(t)$ | 3D position in camera animation $\mathbf{q}^i_j \in \mathbb{Q}^i$, at time $t \in H^i$ |
| $\mathbf{q}^i_{start}$ | Starting camera position. $\mathbf{q}^i_j(t_i) = q^i_{start}, \, \forall \, (i,j)$ |
| $\mathbf{q}^i_{goal}$ | Goal camera position. $\mathbf{q}^i_j(t_i + h) = q^i_{goal} \in \mathbf{V}^i, \, \forall \, (i,j)$ |
| $\dot{\mathbf{q}}(t)$ | Tangent vector of a camera track at time $t$ |
| $D^i_j(t)$ | The camera view vector at time $t$ |
| $(\mathbf{x}, \mathbf{y})$ | angle between two vectors $\mathbf{x}$ and $\mathbf{y}$ |
| $G(x, \sigma)$ | Gaussian decay, equals to $e^{-x^2/(2\sigma^2)}$ |
| $E(x, \lambda)$ | Exponential decay, equals to $e^{-x/\lambda}$ |

Table 1: Notations used in the paper

can be dynamically recomputed to adapt to the characteristics of the scene topology (cluttered *vs.* open environments) and how this adaptation improves our results (Section 6).

## 4 CAMERA ANIMATION SPACE

We propose the design of a *Camera Animation Space* as a relative local frame defined by an initial camera configuration $\mathbf{q}_{start}$ at time $t_i$ and final camera configuration $\mathbf{q}_{goal}$ at time $t_i + h$ (see Figure 2). This local space defines all the possible camera animations that link $\mathbf{q}_{start}$ at time $t_i$ to $\mathbf{q}_{goal}$ at time $t_i + h$. Our goal is to compute the optimal camera motion within this space considering a number of desired features on the trajectory (*e.g.* smoothness, collision and occlusion avoidance along the camera animation, viewpoint preferences and cinematic properties).

We propose to follow a 3-step process: (i) anticipate the target's behavior (*i.e.* its next positions) within a given time horizon, (ii) choose a goal camera viewpoint from which to view the target at the end of the time horizon, and (iii) given this goal viewpoint, and the current one, build and evaluate the space of possible camera animations between them using our camera animation space.

### 4.1 Anticipating the target behavior

We here make a strong assumption that we can anticipate the next positions of the tracked target within a time horizon $H^i$. This is classical in character animation engines in order to decide which animations to trigger (*e.g.* motion matching). We consider $H^i$ begins at time $t_i$ and has a constant user-defined duration of $h$ seconds. Moreover, we consider that the target's behavior will be consistent over the whole horizon $H^i$ (a behaviour being a motion among walking, running, turning). In our implementation, we consider the target as a rigid body (*e.g.* a capsule) with a current speed, acceleration, and behavior. We then simulate the target's motion over time horizon $H^i$ (avoiding collisions with obstacles) and store all simulated positions over time. We rely on PhysicsScene tool from Unity to perform the physical simulation. With this anticipation, we account for the scene geometry which might influence future user inputs. We then refer to the anticipated positions as the target simulated behavior, expressed in the form of a 3d animation curve $B^i(t)$ with $t \in H^i$ (see Figure 3). Note that one may use another technique to anticipate the target behavior. As long as it outputs a 3d animation curve over time, it will not change the overall workflow of our camera system.

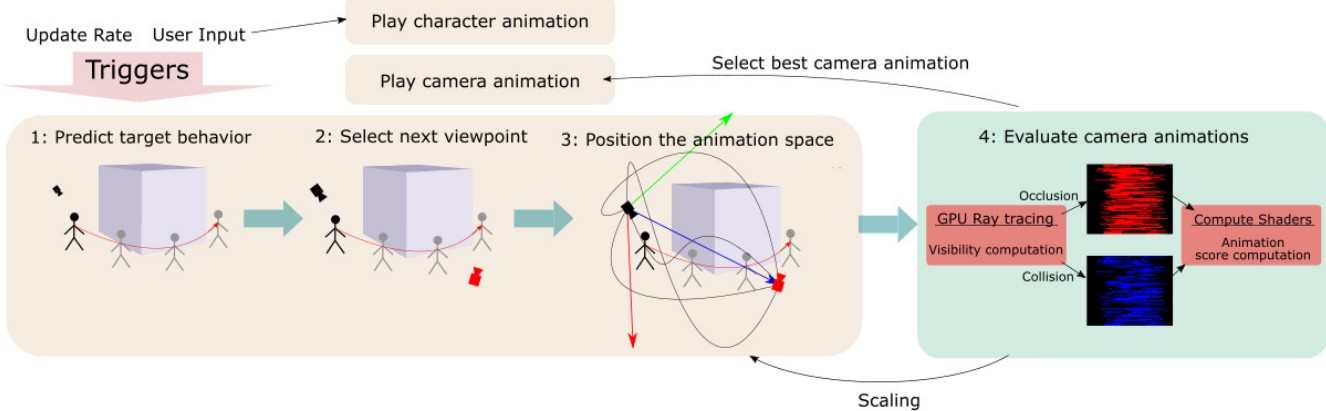

Figure 1: System overview: the orange box represents the CPU part of the system; the green box represent the GPU part of the system

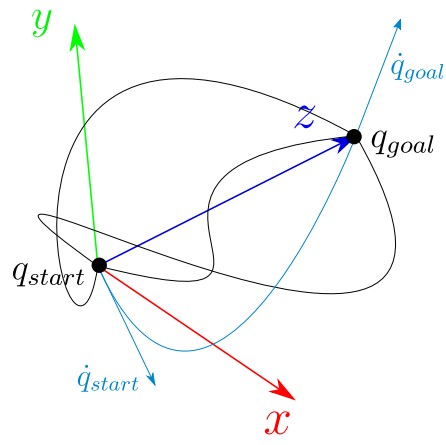

Figure 2: Representation of our *animation space* and its local transform

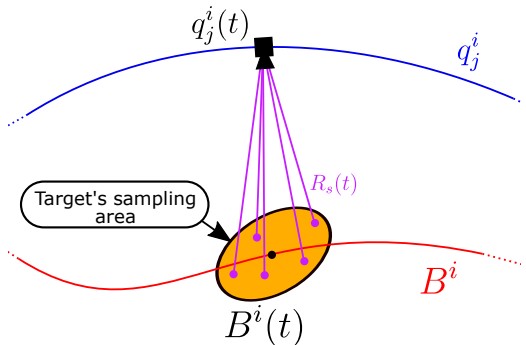

Figure 4: Ray launched from the camera toward the target's sampling area at time t

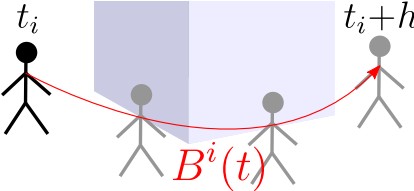

Figure 3: Representation of the target's behaviour curve at iteration $i$

### 4.2 Selecting a goal viewpoint

We now make a second assumption that the user defines a set of viewpoints to portray the target object. By default, one might use a list of stereotypical viewpoints in movies such as 3-quarter front and back views, side views, or bird eye views. These viewpoints are sorted by order of preference in a priority queue $\mathbf{V}$ (order can be fixed by the user, randomly chosen, or scene dependent). Each viewpoint is defined as a 3d position in spherical coordinates $(d, \phi, \theta)$, in the local frame of the target's configuration, where $(\phi, \theta)$ defines the vertical and horizontal viewing angles, and $d$ the viewing distance.

Considering all viewpoints are in $\mathbf{V}$, we pop viewpoints by order of priority. We propose to stop as soon as a viewpoint is promising enough, *i.e.* at time $t_i + h$ neither the target will be occluded from this viewpoint, nor the camera will be in collision with the scene

geometry. We then refer to this selected viewpoint as the goal viewpoint $\mathbf{q}_{goal}$.

Knowing the current camera viewpoint $\mathbf{q}_{start}$ (at time $t_i$) and this goal viewpoint $\mathbf{q}_{goal}$ (at time $t_i + h$), we can define our camera animation space that we further sample and evaluate to select the optimal camera animation.

### 4.3 Sampling camera animations

Given the target behavior to track represented as a curve $B^i(t)$ and the two key viewpoints $\mathbf{q}_{start}$ and $\mathbf{q}_{goal}$, we propose to sample a large set of camera animations between the key viewpoints. We will hereafter note this stochastic set of camera animations as $\mathbb{Q}^i$, and a sampled camera animation as $\mathbf{q}_j^i$, where $j$ is the sample index.

Two requirements should be considered on this sampled space: (i) sampled camera animations should be as-smooth-as-possible, *i.e.* with low jerk, and (ii) the sampled animation space should enforce continuity between successive horizons. To do so, we propose to encode each sampled camera animation as a cubic spline curve on all 3 camera position parameters, as they offer $C^3$ continuity between key-viewpoints. In practice, we make use of Hermite curves which eases the sampling by randomly selecting tangent vectors to the spline curve at start and end positions. $C^1$ continuity between successive Hermite curve portions is enforced by aligning both positions and tangents at connecting positions.

In practice, we propose for each camera animation to complement the starting and the goal camera positions $\mathbf{q}_{start}$ and $\mathbf{q}_{goal}$ by two tangents, *i.e.* the camera velocities $\dot{\mathbf{q}}_{start}$ and $\dot{\mathbf{q}}_{goal}$ (figure 2). To offer a good coverage of the whole *animation space* , we use a

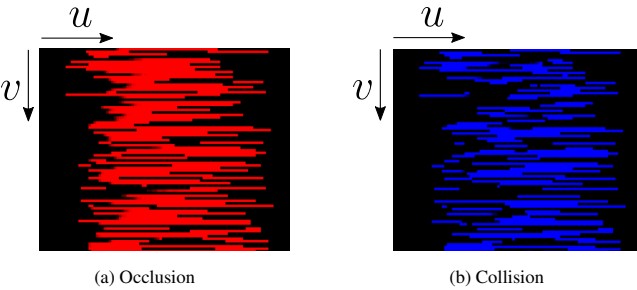

(a) Occlusion          (b) Collision

Figure 5: Example of a part of a Visibility data encoding texture; Black = the target is visible from the camera, Red = the target is occluded or partially occluded from the camera, Blue = the camera is inside the scene geometry.

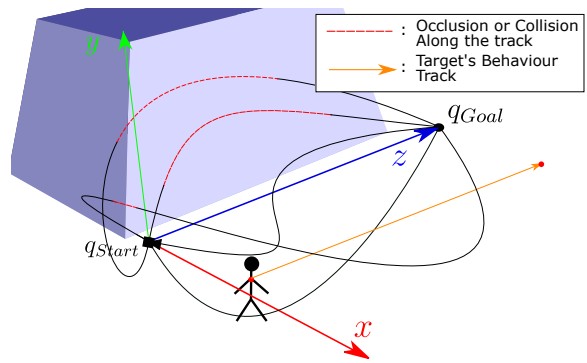

Figure 6: Representation of the positioned *animation space* where parts of some trajectories collide with the scene geometry.

uniform sampling of these tangents in a sphere of radius $r$ (in our tests, we used $r = 5$). The number of sampled animations is left as a user-defined parameter. An evaluation of results for different values is provided in Section 7.2.

The frequent recomputation of the tangent sampling and camera path construction has two drawbacks: its computational expense and the lack of stability over time. To avoid the recomputation, we propose to precompute a graph of uniformly sampled camera animations, in an orthonormal coordinate system (as illustrated in figure 2). In this system, $\mathbf{q}_{start}$ and $\mathbf{q}_{goal}$ have coordinates $(0,0,0)$ and $(0,0,1)$ respectively. Then, for any horizon $H^i$, we apply a proper $4 \times 4$ transform matrix $M^i$ to align the graph onto the computed viewpoints $\mathbf{q}^i_{start}$ and $\mathbf{q}^i_{goal}$. It is worth noting that in $M^i$ the 3d translation, 3d rotation and the scaling on the $z$ axis will lead this axis to match the vector $(\mathbf{q}^i_{goal} - \mathbf{q}^i_{start})$. Two parameters remain free: the scaling for the other two axes ($x$ and $y$). As a first assumption we could use the same scaling as for $z$. However, we will further explain how to choose a better scaling in section 6, in order to adapt the sampled space to the scene geometry.

## 5    EVALUATING CAMERA ANIMATIONS

In this first stage, we have computed a set of camera animations $\mathbb{Q}^i$, that can portray the target objects' behavior within time horizon $H^i$. We now need to select one of these animations as the one to apply to the camera.

### 5.1   Evaluating camera animation quality

Our second stage is devoted to evaluating of the quality of all animations and selecting the most promising one in an efficient way. In the following, we will first detail our evaluation criteria, before focusing on how we perform this evaluation. A camera animation

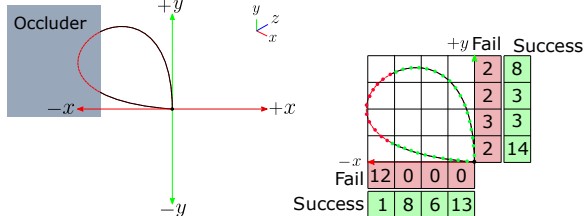

(a) 2D view of an occluded track from the *animation space*    (b) Computation of the fail and success histogram on each axis

Figure 7: Projection of the success and fail of one camera animation on the four axis of resolution $R = 4$. a) Collision and occlusion detection b) Enumeration and projection of the success and fail samples on the axis.

that portrays the motions of a target object should follow a number of requirements, among which the most important are: avoid collisions with the scene and enforce visibility on the target object, while offering a smooth series of intermediate viewpoints to the viewer. To evaluate how well these requirements are satisfied along a camera animation $\mathbf{q}^i_j$, we propose to rely on a set of costs $C_k(t) \in [0,1]$:

**Occlusions and Collisions**    To evaluate how much the target object is occluded from a camera position $q^i_j(t)$, we rely on ray casting as recommended by [12]. We first approximate the target's geometry with a simple abstraction (in our case it is a sphere of the size of the character's upper body). Second we sample a set of points $s \in [0,N]$ on this abstraction, which we position at the object's anticipated position $B^i(t)$. Third, we launch a ray from the camera to each point $s$ (see figure 4). We note $R_s(t)$ the result of this ray cast. We use the same ray to also evaluate if the camera is in collision (*i.e.* inside another object of the scene), by setting its value as:

$$R_s(t) = \begin{cases} 0 & \text{if Visible} \\ 1 & \text{if Occluded} \\ 2 & \text{if Collided} \end{cases} \quad (1)$$

We distinguish a collision from a simple occlusion as follows. By looking at the normal at the hit geometry, we know if the ray has hit a back face or a front face. When the ray hits a back face, $q^i_j(t)$ must be inside a geometry, hence we consider it as a camera collision. Conversely, when the ray hits a front face, $q^i_j(t)$ must be outside a geometry. If the ray does not reach $s$, we consider $s$ as occluded, otherwise we consider it as visible.

Knowing $R_s(t)$, we define our collision and occlusion costs as:

$$C_o(t) = \frac{1}{N} \sum_{s=0}^{N} \begin{cases} 1 & \text{if } R_s(t) = 1 \\ 0 & \text{Otherwise} \end{cases} \quad (2)$$

and

$$C_c(t) = \frac{1}{N} \sum_{s=0}^{N} \begin{cases} 1 & \text{if } R_s(t) = 2 \\ 0 & \text{Otherwise} \end{cases} \quad (3)$$

In our tests, we used $N = 20$.

**Minimizing visual changes**    A smooth camera motion is a motion that avoids sudden changes in visual properties (distance to target and angle to target). We therefore propose an addition metric to evaluate how much the viewpoint changes over time. We split this evaluation into two distinct costs: one on the camera view angle, and one on the distance to the target object. Costs are evaluated for each time step $\delta t$.

Let us denote $D^i_j(t)$ the view vector connecting the target object to the camera computed as:

$$D^i_j(t) = B^i(t) - q^i_j(t)$$

From this view vector, we define the view angle change as:

$$C_{\Delta_{\phi,\theta}}(t) = \frac{(D_j^i(t), D_j^i(t+\delta t))}{\pi} \qquad (4)$$

In a way similar, we propose to rely on a squared distance variation, defined as:

$$\Delta d(t) = (\|D_j^i(t)\| - \|D_j^i(t+\delta t)\|)^2$$

We then define a cost on this distance change which we further normalize as:

$$C_{\Delta d}(t) = 1 - E(\Delta d(t), \lambda) \qquad (5)$$

where $E$ is an exponential decay function, for which we set parameter $\lambda$ to $10^{-4}$.

**Preferred range of distances**  One side effect of the above costs is that for large distances, changes on the view angle and distance will be less penalized. In turn, this will favor large camera animations. It is worth noting that, in the same way, placing the camera too close to the target object is also not desired in general. We should then penalize both behaviors. To do so, we propose to introduce a last cost, aimed at favoring camera animations where the camera remains within a prescribed distance range $[d_{min}, d_{max}]$. We formulate this cost as:

$$C_d(t) = \begin{cases} 1 & \text{if } \|D_j^i(t)\| \notin [d_{min}, d_{max}] \\ 0 & \text{otherwise} \end{cases} \qquad (6)$$

In our tests, we used $[d_{min}, d_{max}] = [0.4, 1]$.

## 5.2  Selecting a camera animation

In a first step, we define the total cost of a camera animation as a weighted sum of its single-criteria costs integrated over time:

$$C = \sum_k w_k \cdot \left[ \int_{t_i}^{t_i+h} C_k(t) \, G(t-t_i, \sigma) \, dt \right] \qquad (7)$$

where $w_k \in [0,1]$ is the weight of criterion $k$. $G$ is a Gaussian decay function, where we set standard deviation $\sigma$ to the value of $h/4$. We also slightly tune the decay to converge towards 0.25 (instead of 0). This way, we give a higher importance to the costs of the beginning of the animation, yet still considering the end. Indeed, as in motion-predictive control, our assumption is that the camera will only play the first part of it (10% in our tests), while the remaining part still brings a long term information on what could be a good camera path. In our tests, typical weights are $w_o = 0.4$, $w_c = 0.2$, $w_d = 0.12$, $w_{\Delta_{\phi,\theta}} = 0.04$, $w_{\Delta d} = 0.04$. We compute the total cost for any camera animation $q_j^i \in \mathbb{Q}^i$ by discretizing the time integral (details are given in the next section). We hereafter refer to this total cost as $C_j^i$.

In a second step, we propose to choose the most promising camera animation for time horizon $H^i$, denoted as $\mathbf{q}^i$, as the one with minimum total cost, *i.e.* :

$$\mathbf{q}^i = \arg\min_j C_j^i \qquad (8)$$

## 5.3  GPU-based evaluation

We have presented our evaluation metric on camera animations. However, some costs are expensive to compute. In particular, occlusion and collision testing requires to trace a large number of rays (*i.e.* $N$ rays, for many time steps, for hundreds of camera animations). It is worth noting that many of our computations are independent and can therefore be performed in parallel. In a similar way, the evaluation of a cost at discretized time steps along a given animation

are also independent. Hence, we propose to cast our evaluation of single costs into a massively-parallel computation on GPU.

We design our system in a way the we only need to send the *animation space* (in orthonormal coordinate system) once to the GPU. Then, when we need to reposition the camera animation space, for horizon $H^i$, we simply update the $4 \times 4$ transform matrix $M^i$. And, from this data, one can straightforwardly compute any camera position $\mathbf{q}_j^i(t)$ for any time $t$.

Second, for occlusion and collision computations, we propose to rely on the recent RTX technology allowing to perform real-time ray casting requests on GPU. To discretize the time integral of our costs, we run $\frac{h}{\delta t}$ kernels per track (each kernel corresponds to a discretized time step). Each kernel launches $N$ rays, one per sample $s$ picked onto the target object. In our tests, we use $N = 20$, and 100 kernels per camera animation. In turn, $\delta t = \frac{h}{100} = 0.05$. The result of these computations are stored into a 2D texture (as shown in figure 5), where the texture coordinates $u$ and $v$ map to one time step $t$ and one animation of index $j$, respectively. Occlusion and collision costs are stored into two different channels.

Third, we rely on a compute shader to compute all other costs, and combine them with occlusion and collision costs. This shader uses one kernel per camera animation. It stores the total cost of all animations into a GPU buffer, finally sent back to CPU where we perform the selection step.

## 6  DYNAMIC TRAJECTORY ADAPTATION

Until now, we have considered a nominal situation where we evaluate the *animation space* and select one camera animation for one given time horizon $H^i$. We now need to consider two other requirements. First, the camera should be animated to track the target object for an unknown duration, larger than $h$. Changes in the target behavior may also occur, due to interactive user inputs. Second, for any horizon $H^i$, some camera animations could be in collision with the scene, or the target could be occluded. This would prevent finding a proper animation to apply. In other words, the space of potential camera animations should be influenced by the surrounding scene geometry. Hereafter, we explain how we account for these requirements.

## 6.1  User inputs and interactive update

We here assume the camera is currently animated along a curve $\mathbf{q}^i$. We then need to compute a new camera animation for a time horizon $H^{i+1}$ in two cases. First, when the target's behavior is changed (*i.e.* the user input has changed). This event indeed breaks the validity of the currently played animation for future time steps. Second, it appears reasonable to consider the target behavior, collision and occlusion information will be less and less reliable as time advances. In a way similar to motion-predictive control, we then compute an animation for an anticipated horizon of length $h$, but only play the first steps, to account for possible dynamic collisions and occlusions. The duration of these first steps is specified as a user-defined ratio of progress along animation $\mathbf{q}^i$. In our tests we used a horizon length $h = 5$ seconds and a ratio of progress of 10%. In turn, the new horizon generally starts at $t_{i+1} = t_i + 0.1h$, while we set $\mathbf{q}_{start}^{i+1} = \mathbf{q}^i(t_{i+1})$.

When an update is required (behavior change, or ratio of progress reached), we recompute a new camera animation for the next horizon $H^{i+1}$: we select a new goal viewpoint (*i.e.* $\mathbf{q}_{goal}^{i+1}$) and update the transform matrix (*i.e.* $M^{i+1}$) to position the camera animation space $\mathbb{Q}^{i+1}$. We then evaluate all camera animations in $\mathbb{Q}^{i+1}$.

**Animation transitions**  . To enforce continuity between animation $q^i$ and the animation $\mathbf{q}^{i+1}$ that is to be selected we rely on an additional cost designed to favor a smooth transition between consecutive animations. This cost penalizes abrupt changes when transitioning between two camera animation curves. Our idea is

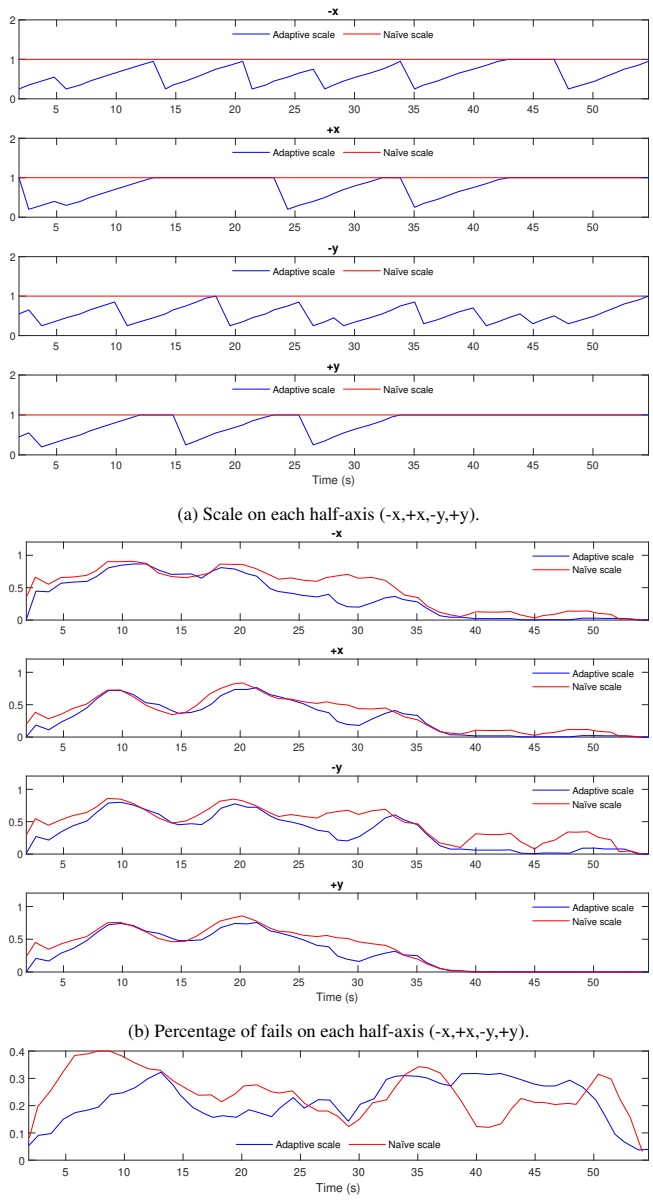

(a) Scale on each half-axis (-x,+x,-y,+y).

(b) Percentage of fails on each half-axis (-x,+x,-y,+y).

(c) Best animation score.

Figure 8: Comparison of our system with an adaptive scale, or with a naïve scale, applied on the camera animation space.

to penalize a wide angle between the tangent vector to camera animation $\mathbf{q}^i$ and the tangent vector to animation $\mathbf{q}_j^{i+1} \in \mathbb{Q}^{i+1}$, at connection time $t_{i+1}$. We write this cost as:

$$C_{i,i+1}(j) = \frac{(\dot{\mathbf{q}}^i(t_{i+1}), \dot{\mathbf{q}}_j^{i+1}(t_{i+1}))}{\pi} \qquad (9)$$

We then rewrite the selection of camera animation $q^{i+1}$ as:

$$\mathbf{q}^{i+1} = \arg\min_j \left[ C_j^{i+1} + w_{i,i+1} \, C_{i,i+1}(j) \right] \qquad (10)$$

where $w_{i,i+1}$ is the relative weight of the transition cost with regards to other costs in our test we use $w_{i,i+1} = 0.2$.

## 6.2 Adapt to scene geometry

Different scene geometries obviously impose different constraints on the camera animations and a single sampled camera animation

space may not enough to tackle all situations. For example, cluttered environment such as corridors would ideally require dedicated samples. In fact, our camera animation space should exhibit as few collisions and occlusions as possible, while still covering as much as possible the free space between the target behavior and the scene geometry. To address this problem, rather than recomputing a new sampling of camera trajectories dedicated to these specific situations, we propose a technique to dynamically adapt our camera animation space to the scene geometry.

To do so, while we evaluate the quality of camera animations for an horizon $H^i$, we analyse how much collisions and occlusions occur. This informs us if the free space is well covered or not. We then propose to dynamically rescale the camera animation space to make it grow or shrink in the next time horizon $H^{i+1}$. This rescaling applies when we update the transform matrix $M^{i+1}$, and on the $x$ and $y$ axes only. It is worth noting that the free space might not be symmetrical around the target behavior (as illustrated figure 6 where the free space is larger on the left than on the right of the target). The same applies to the free space above or below the target. Consequently our idea is to compute four scale values, on all four directions $\{-x, +x, -y, +y\}$ along the axis of the camera animation space. For any camera position along a camera animation, we then apply either two of them, depending on the sign of the position's $x$ and $y$ coordinates in the non-transformed animation space.

To compute this scaling we first leverage the occlusions and collisions evaluation to store additional information: we count fails and successes along each axis. We consider a launched ray along a camera animation (*i.e.* from the camera position at a given time step) as a fail if it is marked as occluded or collided, and as a success if not. Second, we store this information in height arrays: for each half-axis (*e.g.* $+x$ or $-x$), we count successes in one array, and fails in another array. We further discretize this half-axis by using a given resolution $R$ and output two histograms of fails and successes (as illustrated in figure 7). Note that $R$ here defines the scale precision on each axis. At last we use both histograms to compute the new scale to apply. We compute the indices $i_f$ and $i_s$ of the medians of both arrays (fails and successes, respectively). By comparing them, we define how much we should rescale animations along this half-axis. If $i_s < i_f$, we consider that there are too many fails, and multiply the current scale by $i_f/R$ to shrink animations. Otherwise, we consider the free space is not covered enough, and apply a passive inflation to the current scale. The aim of this inflation is to help return to a maximum scale value, when the surrounding geometry allows for large camera animations.

## 7 IMPLEMENTATION AND RESULTS

### 7.1 Implementation

We implemented our camera system within the Unity3D 2019 game engine. We compute our visibility and occlusion textures through raytracing shaders provided with Unity's integrated pipeline and perform our scores for all sampled animations and timesteps through Unity Compute Shaders. All our experimentations (detailed in section 7.2) have been performed on a laptop computer with a Intel Core i9-9880H CPU @ 2.30GHz and a NVIDIA Quadro RTX 4000.

### 7.2 Results

We split our evaluation into three parts. We first validate our adaptive scale mechanism. We then evaluate the robustness of our system, by comparing its performances when using a different number or set of reference camera animations. We finally validate the ability of our system mixing local and global planning approaches to outperform a purely local camera planning system. To do so, we compare results obtained with our system and the one of Burg *et al.* [1], on the same test scenes.

To validate our adaptive scale, we study its impact on the quality of the animation space. For the other evaluations, we compare cam-

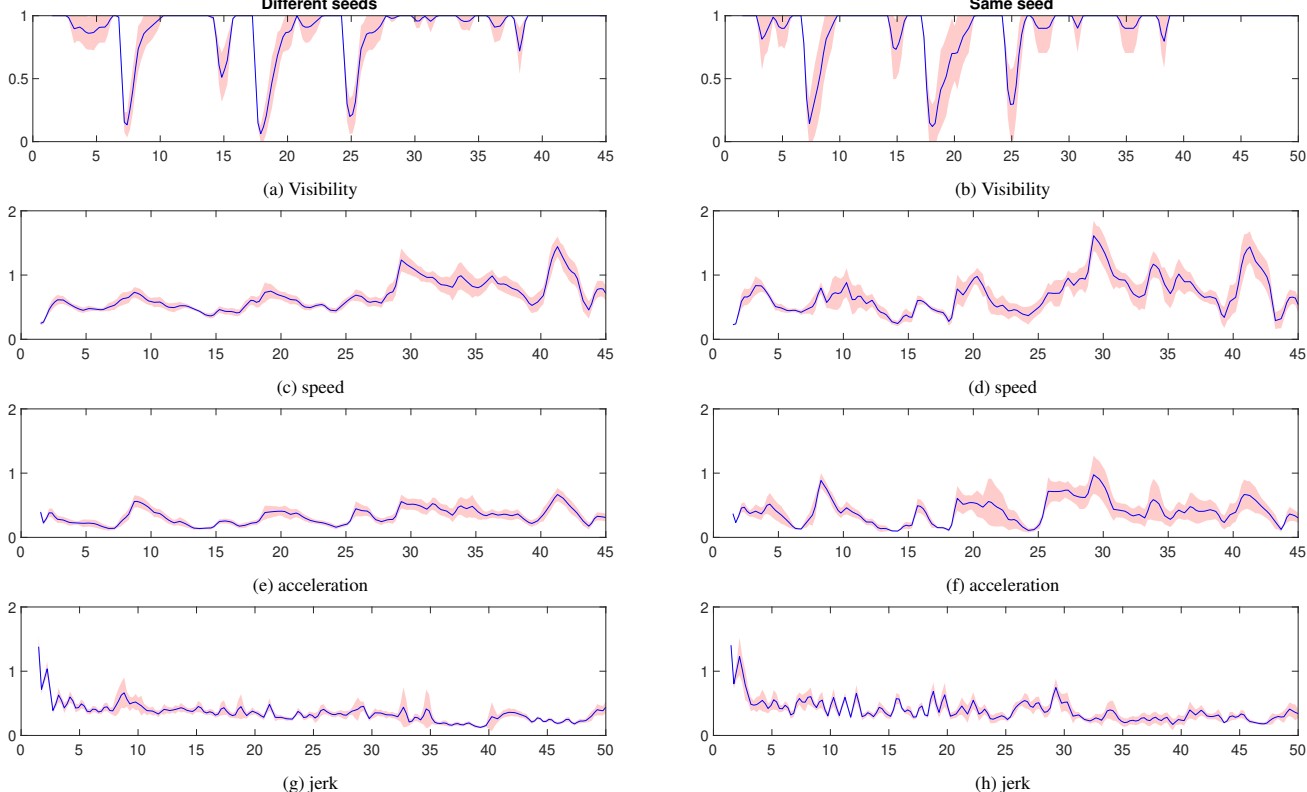

Figure 9: Results for multiple runs, each using a randomly generated camera animation space. This space is sampled with uniform distribution, with 2400 sample camera animations (Hermite curves). Each plot shows the mean value over time (blue), with a 95% confidence interval (red). Left: results for 22 runs using different seeds. Right: results for 10 runs using the same seed.

era systems with regards to two main criteria: how much the camera maintains the visibility on the target object and how smooth camera motions are. We compute visibility by launching rays onto the target object, and calculate the ratio of rays reaching the target. A ratio of 1 (respectively 0) means that the target is fully visible (respectively fully occluded). When relevant, we additionally provide statistics on the duration of partial occlusions. We then compare the quality of camera motions through their time derivatives (speed, acceleration and jerk), which provide a good indication of motion smoothness.

Our comparisons have been performed within 4 different scenes (illustrated in the accompanying video). We validated our system by using (i) a **Toy example scene** where the target is travelling through a maze containing several tight corridors with sharp turns, an open area inside a building, and a ramp. We then performed the comparisons with the technique of Burg *et al.* [1], by using two static scenes and a dynamic scene, which the target goes through: (ii) a scene with a set of columns and a gate (**Columns+Gate**), (iii) a scene with set of small and large spheres (**Spheres**) and (iv) a fully dynamic scene with a set of randomly falling and rolling boxes, and a randomly sliding wall (**Dynamic**). To provide fair comparisons, in the dynamic scene, the random motions of boxes and of the wall are the same for both scenes. In addition, for all tests in a scene, we play a pre-recorded trajectory of the target avatar, but let the camera system run as if the avatar was interactively controlled by a user, to ensure motions are the same in all tests.

### 7.2.1 Impact of adaptive scale

We validate our adaptive scale (section 6.2) by comparing results obtained (i) when we compute and apply the *adaptive scale* on all 4 half-axes $(-x, +x, -y, +y)$, and (ii) when we simply apply the same

scale as for the $z$ axis (which we will call the *naive scale* technique). We ran our tests by using the toy example scene. For each technique, each time we evaluate a new set of camera animations, we output the new scale values and the ratio of fails on each half-axis. In addition, we output and plot the mean cost of the 5 best animations in this set. Results are presented in figure 8.

Figure 8a shows how much our mechanism tightens the animation space (compared to the naive scaling technique) when the avatar is entering corridors, and grows back to the same scale when the avatar reaches less cluttered areas (*e.g.* in the open interior room, or the outdoor area). As expected, our mechanism allows to adapt the scale on half-spaces in a non-symmetrical way. As shown by figure 8b, with our adaptive mechanism, the scaled animation space also exhibits less fails than using the naive scale technique. As shown by figure 8c, it allows finding animations with lower costs most of the time. One exception is between 40s and 50s, where the camera configuration isn't the same because the scale is different. In the naive case the camera is high above the character while in the adaptive case, the camera is closer to the ground, thus the scores is not relevant in this case because the two configurations are too different to be compared.

In the next evaluations, we consider that the adaptive scale mechanism is always activated.

### 7.2.2 Robustness

We study the robustness of our system regarding our randomly generated camera animation space.

In a first step, we evaluate how performances vary if we run our real-time system multiple times on the toy example scene. We also consider two cases: (i) using the same seed for every run (*i.e.* the

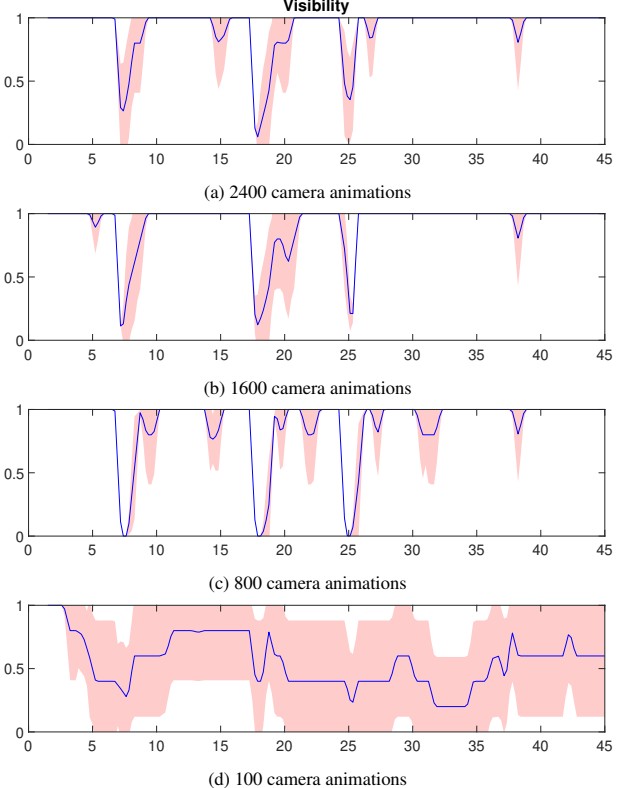

Figure 10: Visibility when varying the number of sampled curves in our camera animation space.

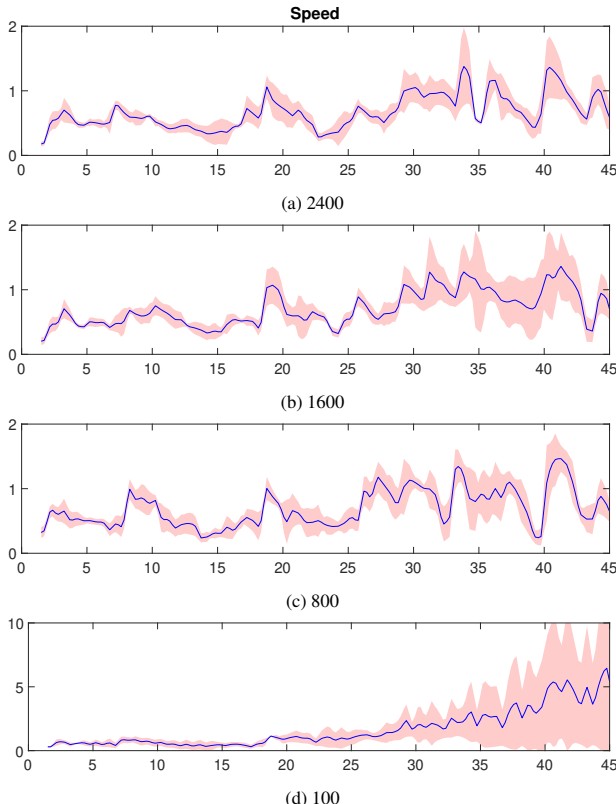

Figure 11: Camera speed when varying the number of sampled curves in our camera animation space.

same animation space is used), and (ii) using a new seed for every run (*i.e.* a new animation space is randomly sampled for each run). For each case, we sample a set of 2400 animations. Results are presented in figure 9. As illustrated, with as many sampled animations, all runs lead to very similar results both on the visibility enforcement and on the camera motion smoothness. Differences are mainly due to variations in the actual framerate of the game engine, hence the rate at which the system takes new decisions.

In a second step, we evaluate how the size of the animation space (*i.e.* the number of sampled animations) impacts performances. We ran our system with 4 different sizes: 2400, 1600, 800 or 100 animations. For each size, we performed 5 runs with random seed, and combined the results in figures 10, 11 and 12. Figures show that lowering the size (at least until 800 animations) still delivers good performances. Our camera system is able to find a series of camera animations maintaining enough visibility on the target object through smooth camera motions. As we expected, for 100 animations, our system's performances are poor: it becomes harder to find animations with sufficient visibility and ensuring smooth camera motions. Obviously as we lower the number of camera animations, the distribution of tangents becomes very sparse, hence breaking our assumption of a uniform sampling. If the sampled *animation space* does not cover enough the free space, it prevents the finding of qualitative animations.

### 7.2.3 Comparison to related work

We also compare our system mixing local and global planning approaches to a purely local camera planning system. We have run our proposed camera system and the local camera planning system of Burg *et al.* [1] in 3 different scenes: two static scenes (**Columns+Gate** and **Spheres**) and a fully dynamic scene

(**Dynamic**). The **Columns+Gate** is the same as in [1] where the avatar is moving between some columns and go through a doorway. In the **Spheres** scene, the avatar is travelling a scene filled with a large set of spheres, which makes it moderately challenging for the camera systems. In the **Dynamic** scene, the avatar must go through a flat area, where a set of boxes are randomly flying, falling, rolling all over the place, and a wall is randomly sliding. This makes it challenging for camera systems to anticipate the scene dynamics and find occlusion-free and collision-free camera paths.

In our camera system, we used 2400 animations, the recomputation rate is set to 0.25s and the adaptive scaling is on. We present results of our tests in figures 13, 14, 15, 16, and 17.

We first compare camera systems along their ability to enforce visibility on the target object (figure 13). Our tests show that for moderately challenging scenes, both lead to relatively good results. Few occlusions occur. However, for a more challenging scene (**Dynamic**), our system outperforms Burg *et al.* 's system. Even if occlusions may occur more often, the degree of occlusion is lower (figure 13b). Moreover, for all 3 scenes, when partial occlusions occur, they are shorter when using our system (figure 13c). This is explained by the fact that when no local solution exist, our system can still find a locally occluded path respecting other constraints, and leading to a less occluded area. This demonstrates our system's ability to better anticipate occlusions especially in dynamic scenes.

Second, we compare the smoothness of camera motions in both camera systems. Figure 14 presents the side-by-side distributions of speed, acceleration and jerk for each system. We also provide the speed, acceleration and jerk along time in figures 15, 16, and 17. One observation we make is that Burg *et al.* 's system leads to lower camera speeds, as it restricts itself to simply following the avatar. In our camera system, the camera is allowed to move faster,

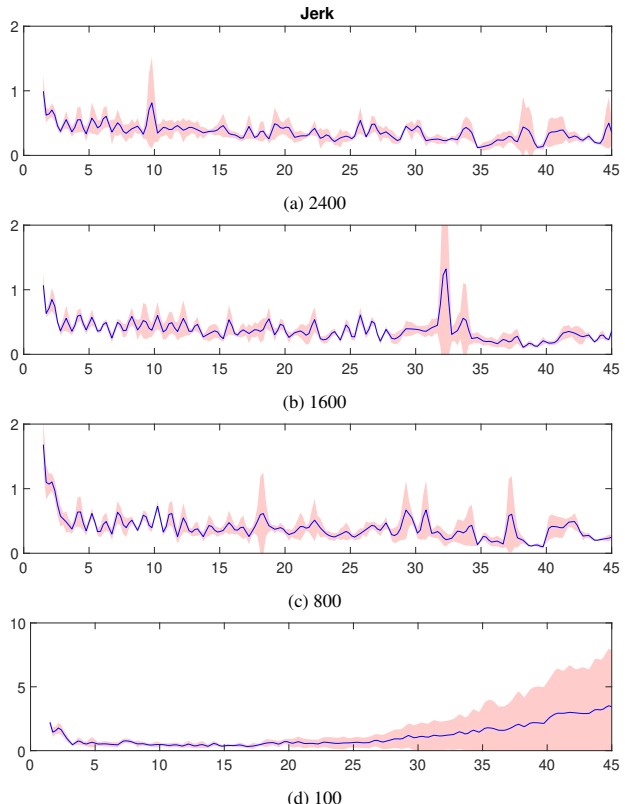

Figure 12: Camera jerk when varying the number of sampled curves in our camera animation space.

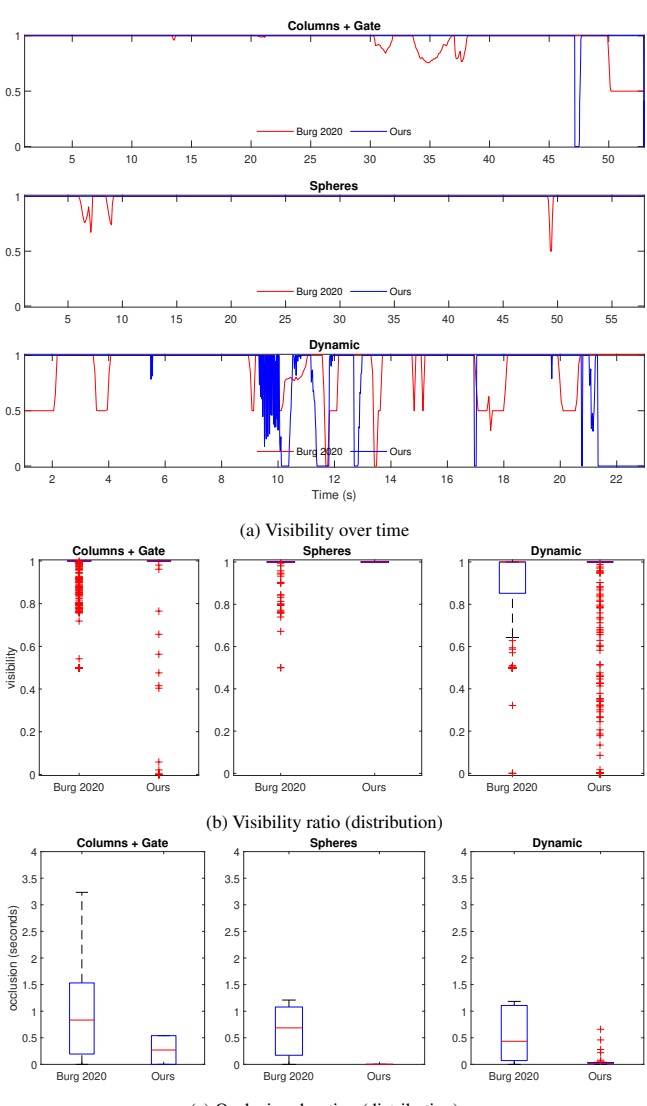

(a) Visibility over time

(b) Visibility ratio (distribution)

(c) Occlusion duration (distribution)

Figure 13: Comparison between our system and Burg *et al.* [1], regarding the target object's visibility (a)(b) and, when not fully visible, the duration of partial occlusion (c).

to bypass the avatar when visibility or another constraint may be poorly satisfied. Yet, our system provides smoother motions (*i.e.* less jerk). One explanation is that local systems often need to steer the camera from local minima (*e.g.* low visibility areas). A side effect is that it may lead, for successive iterations, to an indecision on which direction the camera should take to reach better visibility. In turn, this leads to frequent changes in camera acceleration (hence higher jerk). Conversely, our system has a more global knowledge on the scene, allowing to more easily find a better path, which avoids sacrificing the smoothness of camera motions.

## 8 DISCUSSION AND CONCLUSION

Our system presents a number of limitations. Despite the ability to evaluate thousands of trajectories, strongly cluttered environments remain challenging. As smoothness is enforced, visibility may be lost in specific cases, and designing a techniques that could properly balance between the properties to handle specific cases need to be addressed. Also while the dynamic scale adaptation does improve results by compressing the trajectories in different half spaces, low values in scales prevent the camera from larger motions where necessary. A future work could consist in biasing the sampling in the *animation space* in order to adapt the space to typical local topologies of the 3D environment. Despite the limitations, the proposed work improves over existing contributions by proving an efficient camera tracking technique adapted to dynamic 3D environments and does not require heavy roadmap precomputations.

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
