# OpenReview forum: "Real-Time Cinematic Tracking of Targets in Dynamic Environments"
_graphicsinterface.org/Graphics_Interface/2021/Conference/Second_Cycle — GI 2021_

### Official Review · Reviewer_6wsA · 2021-05-03
**A system for designing camera animation paths in realtime by utilizing moder ray-tracing hardware.**

**Rating:** 6
**Confidence:** 4

**Review:**

**Paper Summary**

This paper proposes a system for generating camera animation to follow and observe a target agent in animated scenes.

The challenges arise from the fact that both agent's behavior and scene contents can change rapidly. Authors propose to overcome these issues by incorporating modern hardware acceleration for ray-tracing queries that are used to evaluate the quality of animation over a large number of specific camera tracks parametrized as Hermite splines.

**Strengths**

Overall, I like the approach and the system design appears sound to me. Parametrizing the search space over a set of splines and interpolating between them is a nice idea and I especially like the clever use of modern ray-tracing hardware to search such space. There are some minor issues that I see with evaluation and clarity, but I remain positive about the paper.

**Notes on evaluation:**

I appreciate the work put in evaluating different scenarios under different metrics, and I acknowledge the challenge of evaluating work like this. I still believe some qualitative evaluation should find its way to the paper - for example comparing the camera behavior in challenging situations for both the comparisons to Burg'20 and the ablation studies.

Comparing the results from the video - in the work by Burg'20, it seems to me like their motion is more stable, much more strongly preferring to follow the target actor in a third-person perspective style. The proposed approach uses a priority queue $V$ to allow the user to specify a preferred type of motion and, from the video, it seems that this choice allows the proposed system to make much more dramatic changes to the camera position to keep its cost metric low. This makes me, as a viewer, slightly prefer the results of Burg'20 in the showcased examples, even
if the visibility metrics are lower.

It would be great to see how the proposed work performs if the camera was made to more strongly adhere to one style of animation, like the aforementioned follow camera.

Given the results, I again qualitatively prefer the results of Burg'20, but I believe that the proposed work could generate comparable frames if the priority queue was limited to follow like camera positions.


**Clarity issues and specific questions:**

- It would be great to have the equations numbered.

- Section 5.1. How the size of the sphere is selected? Since authors do not consider explicit intersections of camera frustum with geometry, the size of this sphere should be somehow tied to the camera frustum / the intrinsic parameters of the camera.

- Section 5.2. Why the equation for total cost contains an integral? The text does not specify how it is discretized, and I do not think that there exists a closed-form solution for this integral.

- Section 5.3. - How is $\delta t$ selected? I can't find in the text how this is selected, it appears to be a parameter of the system?

- Section 6.1 - How is it decided that target behavior has changed? Are there thresholds that are checked when the target has deviated from the predicted path enough?

- Section 6.1 - What is the point of having a scale factor and horizon duration parameters separately, if the two are always multiplied?

**Review Summary**

Overall, while I have issues with some of the choices regarding evaluation and I think the paper could be a bit clearer at times, I still think the strengths outweigh the weaknesses. This is an interesting approach to a complex problem and it is also refreshing to see a clever utilization of new
hardware capabilities.

---

### Official Review · Reviewer_eQrP · 2021-05-03
**Real-Time Cinematic Tracking of Targets in Dynamic Environments**

**Rating:** 6
**Confidence:** 3

**Review:**

The paper presents a method to compute cinematic camera motion for tracking a moving object/character in a 3D environment. This is done essentially by first predicting the tracked object movement over a short period and then searching through a large number of camera paths to choose the 'optimal" path satisfying, continuity, occlusion, collision, view distance, etc. They define a camera space based on camera motion parameters in which the camera paths belong. Occlusion/collision detection is done by casting a large number of rays randomly chosen on a bounding sphere.  For real time performance, ray casting computations are done in the GPU.  The overall methodology is straight forward, in that one does not see too much innovation. overall it is a good effort. One problem that I foresee is that their brute force search for the optimal camera path will not scale to complex 3D environment, where the ray casting operation would be the bottle neck. The other more important issue that cinematic motion probably goes beyond geometric issues such as continuity, occlusion, etc. So, the question is acceptability of the generated camera path by cinematographers. It would have been nice to see some reactions from that community, even if a formal user study may be difficult.

On the writing front, there are quite a few typos and English language/expression problems, which can be easily fixed by giving it a careful reading. I have not checked the formulations thoroughly, but seemed OK to me. However, in once place they cubic splines give C_3 continuity. That is not true in general.  Hermite cubics which they use give C_1 continuity at the junctions. They say that later on. This needs to be checked and corrected if necessary.

---

### Official Review · Reviewer_Mv11 · 2021-05-04
**GPU driven real-time camera control for target tracking and visibility in dynamic virtual environments**

**Rating:** 6
**Confidence:** 4

**Review:**

**Summary**
This paper proposes a methodology for automatically generating camera animations for tracking a target object/character in a dynamic virtual environment. A "shooting" type approach is used to synthesize many different camera animations for a given time horizon, and the best one is selected based on a multi-objective cost function.  The generated camera trajectories are (mostly) free from occlusion and can handle dynamic changes, such as changes in target motion or other moving objects.

**Clarity of exposition**
I did find many small issues with the text (too many to copy here). So I would recommend a thorough editing pass to clean-up grammar and spelling errors. Also, it was a bit of a chore to read some sections of the paper due to the misplacement/overuse of commas.  Please try and limit that.

Also, the use of mathematical notation is confusing in some places, e.g., in Section 4, H is used as a duration, e.g. "t+H", whereas it is later presented as a set, e.g. "t \in H".   Numbered equations would have been helpful.

**Reproducibility**
Most of the technical details are sufficient so that a student/practitioner could produce a reasonable version of the system described in the paper. However, some are ambiguous. For instance, C_deltad uses an "exponential decay function" E that is not well defined (why not explicitly provide the exponential function in the paper?) Similarly for the "Gaussian decay" function used to weight the cost functions over time, just provide it. Regarding weights, the values of w_k for each cost function aren't provided. Other parameters, such as the [dmin, dmax] range, are also missing from the paper.

Also, only a few details are presented about the synthesis of the target behavior during the first step of the pipeline. This seems like a fairly important step. So, further details would be appreciated here, i.e. this step seems to involve a physics simulation and computation of a hermite curve... it doesn't sound trivial!

**Evaluation of the method**
The only comparison provided in the paper and suppl. video is against the method of Burg et al. (2020), which is a local planning approach. Since this paper situates the proposed approach as a local+global hybrid, I would have expected to also see a comparison versus a global planning approach that is capable of handling dyamic occluders (e.g., Oskam et al. 2009)

However, for the most part, the proposed method does produce more interesting transitions and maintains better visiblity than Burg et al.  Although, I personally preferred the results from Burg since the viewpoint remained more consistent thoughout, whereas the proposed approach kept changing the vantage point. I think the comparison here would work better if the proposed approach tried to maintain a similar viewpoint, i.e. from behind and above.

There seem to be occlusions that were missed in the video comparison to Burg. Here are a few instances :
   * 2:34 character is inside an object but still counted as visible
   * 2:36 camera is occluded by a box
   * 2:40 camera is again occluded by a box
Also, the "jerk" and "visibility" plot animations do not seem to be synchronized with the scene animation sometimes.

There seem to be two different scales used for plotting the "jerk", and the Burg plot does not match the values provided in the paper. Based on the video, I am not quite convinced that the Burg approach is orders of magnitude more "jerky" compared to the new approach.

Finally, some sort of ablation study would have been helpful to understand the importance of each cost function (e.g. without vs with the animation transition cost).

**Overall evaluation**
The overall idea to apply model predictive control (MPC) for camera tracking is an interesting one, and it seems to be reasonably well executed here. The use of GPU hardware to evaluate a large number of camera trjectories is especially nice.  The results in the paper demonstrate that the proposed method is able to responsively replan and change the camera trajectory to maintain visiblity of the target.

However, the exposition and evaluation of the approach leave a lot to be desired.  So while I am leaning positive about the paper, I would ask the authors to consider the points raised above to strengthen the work.

---

### Meta-Review · Area_Chair_R649 · 2021-05-06

**Recommendation:** Accept
**Confidence:** 4

**Metareview:**

Dear authors, the reviewers found the proposed methodology for real-time in-game camera control to be well devised and executed.  The resourceful use of GPU hardware to evaluate many different trajectories is also an aspect of the work that is appreciated. In the end, despite concerns about the evaluation of the technique, reviewers were postive about the paper.

However, all reviewers noted that there are issue with the writing and technical clarity of the paper. Please take their feedback into strong consideration when preparing the final version.

---

### Decision · Program_Chairs · 2021-05-08

Accept